# Contrastive Explanations for Reinforcement Learning via Embedded Self Predictions

**Zhengxian Lin, Kim-Ho Lam, Alan Fern**
Department of EECS
Oregon State University
{linzhe, lamki, alan.fern}@oregonstate.edu

## Abstract

We investigate a deep reinforcement learning (RL) architecture that supports explaining why a learned agent prefers one action over another. The key idea is to learn action-values that are directly represented via human-understandable properties of expected futures. This is realized via the embedded self-prediction (ESP) model, which learns said properties in terms of human provided features. Action preferences can then be explained by contrasting the future properties predicted for each action. To address cases where there are a large number of features, we develop a novel method for computing minimal sufficient explanations from an ESP. Our case studies in three domains, including a complex strategy game, show that ESP models can be effectively learned and support insightful explanations.

## 1 Introduction

Traditional RL agents explain their action preference by revealing action $A$ or $B$'s predicted values, which provide little insight into its reasoning. Conversely, a human might explain their preference by contrasting meaningful properties of the predicted futures following each action. In this work, we develop a model allowing RL agents to explain action preferences by contrasting human-understandable future predictions. Our approach learns deep generalized value functions (GVFs) (Sutton et al., 2011) to make the future predictions, which are able to predict the future accumulation of arbitrary features when following a policy. Thus, given human-understandable features, the corresponding GVFs capture meaningful properties of a policy's future trajectories.

To support sound explanation of action preferences via GVFs, it is important that the agent uses the GVFs to form preferences. To this end, our first contribution is the *embedded self-prediction (ESP) model*, which: 1) directly "embeds" meaningful GVFs into the agent's action-value function, and 2) trains those GVFs to be "self-predicting" of the agent's Q-function maximizing greedy policy. This enables meaningful and sound contrastive explanations in terms of GVFs. However, this circularly defined ESP model, i.e. the policy depends on the GVFs and vice-versa, suggests training may be difficult. Our second contribution is the ESP-DQN learning algorithm, for which we provide theoretical convergence conditions in the table-based setting and demonstrate empirical effectiveness.

Because ESP models combine embedded GVFs non-linearly, comparing the contributions of GVFs to preferences for explanations can be difficult. Our third contribution is a novel application of the integrated gradient (IG) (Sundararajan et al., 2017) for producing explanations that are sound in a well-defined sense. To further support cases with many features, we use the notion of minimal sufficient explanation (Juozapaitis et al., 2019), which can significantly simplify explanations while remaining sound. Our fourth contribution is case studies in two RL benchmarks and a complex real-time strategy game. These demonstrate insights provide by the explanations including both validating and finding flaws in reasons for preferences.

**In Defense of Manually-Designed Features.** It can be controversial to provide deep learning algorithms with engineered meaningful features. The key question is whether the utility of providing such features is worth the cost of their acquisition. We argue that for many applications that can benefit from informative explanations, the utility will outweigh the cost. Without meaningful features, explanations must be expressed as visualizations on top of lower-level perceptual information (e.g.

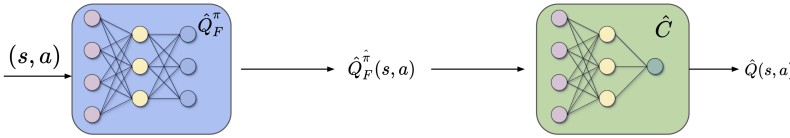

Figure 1: The ESP model provides a estimate of the agent's Q-function for any state-action pair. The model first maps a state-action pair $(s, a)$ to a GVF vector $\hat{Q}_F^{\hat{\pi}}$ of the agent's greedy policy $\hat{\pi}(s) = \hat{Q}(s, a)$. This vector is then processed by the combining function $\hat{C}$, which produces a Q-value estimate $\hat{Q}(s, a)$. The embedded GVF is self-predicting in the sense that it is predicting values of the greedy policy for which it is being used to compute.

saliency/attention maps). Such explanations have utility, but they may not adequately relate to human-understandable concepts, require subjective interpretation, and can offer limited insight. Further, in many applications, meaningful features already exist and/or the level of effort to acquire them from domain experts and AI engineers is reasonable. It is thus important to develop deep learning methods, such as our ESP model, that can deliver enhanced explainability when such features are available.

## 2 EMBEDDED SELF-PREDICTION MODEL

An MDP is a tuple $\langle S, A, T, R \rangle$, with states $S$, actions $A$, transition function $T(s, a, s')$, and reward function $R(s, a)$. A policy $\pi$ maps states to actions and has Q-function $Q^\pi(s, a)$ giving the expected infinite-horizon $\beta$-discounted reward of following $\pi$ after taking action $a$ in $s$. The optimal policy $\pi^*$ and Q-function $Q^*$ satisfy $\pi^*(s) = \arg\max_a Q^*(s, a)$. $Q^*$ can be computed given the MDP by repeated application of the *Bellman Backup Operator*, which for any Q-function $Q$, returns a new Q-function $B[Q](s, a) = R(s, a) + \beta \sum_{s'} T(s, a, s') \max_{a'} Q(s', a')$.

We focus on RL agents that learn an approximation $\hat{Q}$ of $Q^*$ and follow the corresponding greedy policy $\hat{\pi}(s) = \arg\max_a \hat{Q}(s, a)$. We aim to explain a preference for action $a$ over $b$ in a state $s$, i.e. explain why $\hat{Q}(s, a) > \hat{Q}(s, b)$. Importantly, the explanations should be meaningful to humans and soundly reflect the actual agent preferences. Below, we define the embedded self-prediction model, which will be used for producing such explanations (Section 4) in terms of generalized value functions.

**Generalized Value Functions (GVFs).** GVFs (Sutton et al., 2011) are a generalization of traditional value functions that accumulate arbitrary feature functions rather than reward functions. Specifically, given a policy $\pi$, an $n$-dimensional state-action feature function $F(s, a) = \langle f_1(s, a), \ldots, f_n(s, a) \rangle$, and a discount factor $\gamma$, the corresponding $n$-dimensional GVF, denoted $Q_F^\pi(s, a)$, is the expected infinite-horizon $\gamma$-discounted accumulation of $F$ when following $\pi$ after taking $a$ in $s$. Given an MDP, policy $\pi$, and feature function $F$, the GVF can be computed by iterating the *Bellman GVF operator*, which takes a GVF $Q_F$ and returns a new GVF $B_F^\pi[Q_F](s, a) = F(s, a) + \gamma \sum_{s'} T(s, a, s') Q_F(s', \pi(s'))$.

To produce human-understandable explanations, we assume semantically-meaningful features are available, so that the corresponding GVFs describe meaningful properties of the expected future—e.g., expected energy usage, or time spent in a particular spatial region, or future change in altitude.

**ESP Model Definition.** Given policy $\pi$ and features $F$, we can contrast actions $a$ and $b$ via the GVF difference $\Delta_F^\pi(s, a, b) = Q_F^\pi(s, a) - Q_F^\pi(s, b)$, which may highlight meaningful differences in how the actions impact the future. Such differences, however, cannot necessarily be used to soundly explain an agent preference, since the agent may not explicitly consider those GVFs for action selection. Thus, the ESP model forces agents to directly define action values, and hence preferences, in terms of GVFs of their own policies, which allows for such differences to be used soundly.

As depicted in Figure 1, the ESP model embeds a GVF $Q_F^{\hat{\pi}}$ of the agent's greedy policy $\hat{\pi}$ into the agents Q-function $\hat{Q}$, via $\hat{Q}(s, a) = \hat{C}(\hat{Q}_F(s, a))$, where $\hat{C} : R^n \to R$ is a learned combining function from GVF vectors to action values. When the GVF discount factor $\gamma$ is zero, the ESP model becomes a direct combination of the features, i.e. $\hat{Q}(s, a) = \hat{C}(F(s, a))$, which is the traditional approach to using features for function approximation. By using $\gamma > 0$ we can leverage human-

provided features in a potentially more powerful way. Because an ESP agent represents action-values via GVF components, it is possible to produce sound contrastive explanations in terms of GVFs, as described in Section 4.

In general, the ability to learn a quality Q-function and hence policy using the ESP model requires that the GVF features are sufficiently expressive. While, in concept, using a single feature equal to the reward is sufficient for learning the Q-function (i.e. the GVF is the Q-function and the identity combining function could be used), that choice does not support explainability. Thus, it is desirable to use a set of features that meaningfully decompose important aspects of the environment and at the same time have GVFs that are expressive enough to combine into the Q-function. In Section 6, we describe the generic schema used for GVF features in our experimental environments.

## 3 ESP Model Training: ESP-DQN

We will represent the learned combining function, $\hat{C}$, and GVF, $\hat{Q}_F$, as neural networks with parameters $\theta_C$ and $\theta_F$. The goal is to optimize the parameters so that $\hat{Q}(s, a) = \hat{C}(\hat{Q}_F(s, a))$ approximates $Q^*$ and $\hat{Q}_F(s, a)$ approximates $Q_F^{\pi^*}(s, a)$. The GVF accuracy condition is important since humans will interpret the GVF values in explanations. A potential learning complication is the circular dependence where $Q_F^{\hat{\pi}}$ is both an input to $\hat{Q}$ and depends on $\hat{Q}$ through the greedy policy $\hat{\pi}$. Below we overview our learning algorithm, *ESP-DQN*, a variant of DQN (Mnih et al., 2015), which we later show to be empirically effective. Full pseudo-code is provided in the Appendix A.

ESP-DQN follows an $\epsilon$-greedy exploration policy while adding transitions to a replay buffer $D = \{(s_i, a_i, r_i, F_i, s_i')\}$, where $F_i$ is the feature vector for GVF training. Each learning step updates $\theta_C$ and $\theta_F$ using a random mini-batch. Like DQN, updates are based on a *target network*, which uses a second set of *target parameters* $\theta_C'$ and $\theta_F'$, defining target combining and GVF functions $\hat{C}'$ and $\hat{Q}_F'$, yield target Q-function $\hat{Q}'(s, a) = \hat{C}'(\hat{Q}_F'(s, a))$. The target parameters are updated to the values of the non-target parameters every $K$ learning steps and otherwise held fixed.

**Combination Function Update.** Since the output of $\hat{C}$ should approximate $Q^*$, optimizing $\theta_C$ can use traditional DQN updates. The updates, however, only impact $\theta_C$ while keeping $\theta_F$ fixed so that the GVF output $\hat{Q}_F(s, a)$ is viewed as a fixed input to $\hat{C}$. Given a mini-batch the update to $\theta_C$ is based on L2 loss with a target value for sample $i$ being $y_i = r_i + \beta\hat{Q}'(s_i', \hat{a}_i')$, where $\hat{a}_i' = \arg\max_a \hat{Q}'(s', a)$ is the greedy action of the target network.

**GVF Update.** Training $Q_F^\pi$ is similar to learning a critic in actor-critic methods for the evolving greedy policy, but instead of learning to predict long-term reward, we predict the long-term accumulation of $F$. Given a mini-batch we update $\theta_F$ based on L2 loss at the output of $\hat{Q}_F$ with respect to a target value $y_i = F_i + \gamma\hat{Q}_F'(s_i', \hat{a}_i')$, where $\hat{a}_i$ is the same target greedy action from above.

**Convergence.** Even with sufficiently expressive features, most combinations of function approximation and Q-learning, including DQN, do not have general convergence guarantees (Sutton & Barto, 2018). Rather, for table-based representations that record a value for each state-action pair, Q-learning, from which DQN is derived, almost surely converges to $Q^*$ (Watkins & Dayan, 1992), which at least shows that DQN is built on sound principles. We now consider convergence for ESP-Table, a table-based analog of ESP-DQN.

ESP-Table uses size 1 mini-batches and updates target tables (i.e. analogs of target networks) every $K$ steps. The $\hat{Q}_F$ table is over state-action pairs, while for $\hat{C}$ we assume a hash function $h$ that maps its continuous GVF inputs to a finite table. For example, since GVFs are bounded, this can be done with arbitrarily small error via quantization. A pair of feature and hash function $(F, h)$ must be sufficiently expressive to provide any convergence guarantee. First, we assume $h$ is *locally consistent*, meaning that for any input $q$ there exists a finite $\epsilon$ such that for all $|q' - q| \le \epsilon$, $h(q) = h(q')$. Second, we assume the pair $(F, h)$ is *Bellman Sufficient*, which characterizes the representational capacity of the $\hat{C}$ table after Bellman GVF backups (see Section 2) with respect to representing Bellman backups.

**Definition 1** (Bellman Sufficiency). *A feature and hash function pair $(F, h)$ is* Bellman sufficient *if for any ESP model $\hat{Q}(s, a) = \hat{C}(\hat{Q}_F(s, a))$ with greedy policy $\hat{\pi}$ and state-action pairs $(s, a)$ and $(x, y)$, if $h(\hat{Q}_F^+(s, a)) = h(\hat{Q}_F^+(x, y))$ then $B[\hat{Q}](s, a) = B[\hat{Q}](x, y)$, where $\hat{Q}_F^+ = B_F^{\hat{\pi}}[\hat{Q}_F]$.*

Let $\hat{C}^t$, $\hat{Q}_F^t$, $\hat{Q}^t$, and $\hat{\pi}^t$ be random variables denoting the learned combining function, GVF, corresponding Q-function, and greedy policy after $t$ updates. The following gives conditions for convergence of $\hat{\pi}^t$ to $\pi^*$ and $\hat{Q}_F^t$ to a neighborhood of $Q_F^*$ given a large enough update interval $K$.

**Theorem 1.** *If ESP-Table is run under the standard conditions for the almost surely (a.s.) convergence of Q-learning and uses a Bellman-sufficient pair $(F, h)$ with locally consistent $h$, then for any $\epsilon > 0$ there exists a finite target update interval $K$, such that for all $s$ and $a$, $\hat{\pi}^t(s)$ converges a.s. to $\pi^*(s)$ and $\lim_{t \to \infty} |\hat{Q}_F^t(s, a) - Q_F^*(s, a)| \leq \epsilon$ with probability 1.*

The full proof is in the Appendix B. It is an open problem of whether a stronger convergence result holds for $K = 1$, which would be analogous to results for traditional Q-learning.

## 4    CONTRASTIVE EXPLANATIONS FOR THE ESP MODEL

We focus on contrastive explanation of a preference, $\hat{Q}(s, a) > \hat{Q}(s, b)$, that decomposes the preference magnitude $\hat{Q}(s, a) - \hat{Q}(s, b)$ in terms of components of the GVF difference vector $\Delta_F(s, a, b) = \hat{Q}_F(s, a) - \hat{Q}_F(s, b)$. Explanations will be tuples $\langle \Delta_F(s, a, b), W(s, a, b) \rangle$, where $W(s, a, b) \in R^n$ is an attribution weight vector corresponding to $\Delta_F(s, a, b)$. The meaningfulness of an explanation is largely determined by the meaningfulness of the GVF features. We say that an explanation is sound if $\hat{Q}(s, a) - \hat{Q}(s, b) = W(s, a, b) \cdot \Delta_F(s, a, b)$, i.e. it accounts for the preference magnitude. We are interested in explanation methods that only return sound explanations, since these explanations can be viewed as certificates for the agent's preferences. In particular, the definition implies that $W(s, a, b) \cdot \Delta_F(s, a, b) > 0$ if and only if $\hat{Q}(s, a) > \hat{Q}(s, b)$. In the simple case of a linear combining function $\hat{C}$ with weights $w \in R^n$, the preference magnitude factors as $\hat{Q}(s, a) - \hat{Q}(s, b) = w \cdot \Delta_F(s, a, b)$. Thus, $\langle \Delta_F(s, a, b), w \rangle$ is a sound explanation for any preference.

**Non-Linear Combining Functions.** Non-linear combining functions are necessary when it is difficult to provide features that support good policies via linear combining functions. Since the above linear factoring does not directly hold for non-linear $\hat{C}$, we draw on the *Integrated Gradient (IG)* (Sundararajan et al., 2017), which was originally developed to score feature importance of a single input relative to a "baseline" input. We adapt IG to our setting by treating the less preferred action as the baseline, which we describe below in the terminology of this paper.

Let $X_{sa} = \hat{Q}_F(s, a)$ and $X_{sb} = \hat{Q}_F(s, b)$ be the GVF outputs of the compared actions. Given a differentiable combining function $\hat{C}$, IG computes an attribution weight $\theta_i(s, a, b)$ for component $i$ by integrating the gradient of $\hat{C}$ while interpolating between $X_a$ and $X_b$. That is, $\theta_i(s, a, b) = \int_0^1 \frac{\partial \hat{C}(X_{sb} + \alpha \cdot (X_{sa} - X_{sb}))}{\partial X_{sa,i}} d\alpha$, which we approximate via finite differences. The key property is that the IG weights linearly attributes feature differences to the overall output difference, i.e. $\hat{C}(X_{sa}) - \hat{C}(X_{sb}) = \theta(s, a, b) \cdot (X_{sa} - X_{sb})$. Rewriting this gives the key relationship for the ESP model.

$$\hat{Q}(s, a) - \hat{Q}(s, b) = \hat{C}(\hat{Q}_F(s, a)) - \hat{C}(\hat{Q}_F(s, b)) = \theta(s, a, b) \cdot \Delta_F(s, a, b) \tag{1}$$

Thus, $\text{IGX}(s, a, b) = \langle \Delta_F(s, a, b), \theta(s, a, b) \rangle$ is a sound explanation, which generalizes the above linear case, since for linear $\hat{C}$ with weights $w$, we have $\theta(s, a, b) = w$. In practice, we typically visualize $\text{IGX}(s, a, b)$ by showing a bar for each component with magnitude $\theta_i(s, a, b) \cdot \Delta_F(s, a, b)$, which reflects the positive/negative contributions to the preference (e.g. Figure 3a bottom-right).

**Minimal Sufficient Explanations.** When there are many features $\text{IGX}(s, a, b)$ will likely overwhelm users. To soundly reduce the size, we use the concept of minimal sufficient explanation (MSX), which was recently developed for the much more restricted space of linear reward decomposition models (Juozapaitis et al., 2019). Equation 1, however, allows us to adapt the MSX to our non-linear setting. Let $P$ and $N$ be the indices of the GVF components that have positive and negative attribution to the preference, i.e., $P = \{i \; : \; \Delta_{F,i}(s, a, b) \cdot \theta_i(s, a, b) > 0\}$ and $N = \{1, \ldots, n\} - P$. Also, for an arbitrary subset of indices $E$, let $S(E) = \sum_{i \in E} |\Delta_{F,i}(s, a, b) \cdot \theta_i(s, a, b)|$ be the total magnitude of the components, which lets the preference be expressed as $S(P) > S(N)$. The key idea of the MSX is that often only a small subset of positive components are required to overcome negative components and maintain the preference of $a$ over $b$. An MSX is simply a minimal set of such positive components. Thus, an MSX is a solution to $\arg \min \{|E| \; : \; E \subseteq P, S(E) > S(N)\}$, which

is not unique in general. We select a solution that has the largest positive weight by sorting $P$ and including indices into the MSX from largest to smallest until the total is larger than $S(N)$.

## 5 RELATED WORK

Prior work considered linear reward decomposition models with known weights for speeding up RL (Van Seijen et al., 2017), multi-agent RL (Russell & Zimdars, 2003; Kok & Vlassis, 2004), and explanation (Juozapaitis et al., 2019). This is a special case of the ESP model, with GVF features equal to reward components and a known linear combining function. Generalized value function networks (Schlegel et al., 2018) are a related, but orthogonal, model that combines GVFs (with given policies) by treating GVFs as features accumulated by other GVFs. Rather, our GVFs are used as input to a combining network, which defines the policy used for the GVF definition. Integrating GVF networks and the ESP model is an interesting direction to consider.

The MSX for linear models was originally introduced for MDP planning (Khan et al., 2009) and more recently for reward decomposition (Juozapaitis et al., 2019). We extend to the non-linear case. A recent approach to contrastive explanations (Waa et al., 2018) extracts properties from policy simulations at explanation time (Waa et al., 2018), which can be expensive or impossible. Further, the explanations are not sound, since they are not tied to the agent's internal preference computation. Saliency explanations have been used in RL to indicate important parts of input images (Greydanus et al., 2018; Iyer et al., 2018; Gupta et al., 2020; Atrey et al., 2020; Olson et al., 2019). These methods lack a clear semantics for the explanations and hence any notion of soundness.

## 6 EXPERIMENTAL CASE STUDIES

Below we introduce our domains and experiments, which address these questions: 1) (Section 6.2) Can we learn ESP models that perform as well as standard models? 2) (Section 6.2) Do the learned ESP models have accurate GVFs? 3) (Section 6.3) Do our explanations provide meaningful insight?

### 6.1 ENVIRONMENT DESCRIPTION

**Schema for Selecting GVF Features.** Before introducing the environments we first describe the schema used to select GVF features across these environments. This schema can serve as a general starting point for applying the ESP model to new environments. In general, episodic environments have two main types of rewards: 1) a *terminal reward*, which occurs at the end of an episode and can depend on the final state, and 2) *pre-terminal rewards*, which occur during the episode depending on the states and/or actions. Since the value of a policy will typically depend on both types of rewards, it is important to have GVF features that capture both terminal and pre-terminal that are potentially relevant and interpretable. Thus, in each domain, as describe below, we include simple *terminal GVF features* that describe basic conditions at the end of the episode (e.g. indicating if the cart went out-of-bound in Cartpole). In addition, we include *pre-terminal GVF features* that are obtained from the state variables of the environment or derived reward variables that are used to compute the reward function, which are typically readily available from a domain description.

Discrete state or reward variables can simply be encoded as indicator GVF features. For continuous state and reward variables we consider two options: a) When a variable has a small number of meaningful regions, we can use indicator features for the regions as features. The GVFs then indicate how long the agent is in each region; b) We also consider delta GVF features that are equal to the change in a variable across a time step. The GVF value for these features can be interpreted as the future change in the variables' values. While we focus on the above generic GVF features in this paper, an agent designer can define arbitrary GVF features based on their intuition and knowledge.

**Lunar Lander.** We use the standard OpenAI Gym version of Lunar Lander, a physics simulation game where the agent aims to safely land a rocket ship in a target region by deciding at each step which of three thrusters (if any) to activate. The raw state variables are the position and velocity vectors and the reward function penalizes crashing, rewards landing in the goal area, and includes other "shaping" reward variables. These variables are all easily extracted from the simulation environment. In this domain, the continuous variables do not have intuitively meaningful discretizations, so we use the delta features as the main features for explanation case studies (ESP-continuous). However, to illustrate that learning can be done with the discretization approach in this domain, we also

include results for the continuous features being discretized into 8 uniform bins (ESP-discrete). The pre-terminal features are based on the variables distance-to-goal, velocity, tilt-angle, right landing leg in goal position, left landing leg in goal position, main engine use, side engine use. The terminal feature is an indicator of safely landing.

**Cart Pole.** We use the standard OpenAI Gym Cart Pole environment, a physics simulation where the agent aims to vertically balance a free-swinging pole attached to a cart that can have a force applied to the left or right each step. The state variables include the position, pole angle, and their velocities, and reward is a constant +1 until termination, which occurs when either the pole falls below a certain angle from vertical, moves out of bounds, or after 500 steps.

Since the CartPole variables discretize into a small number of intuitively meaningful regions, we consider both discrete (ESP-discrete) and delta encodings (ESP-continuous) of the GVF features. For ESP-discrete, there are 8 pre-terminal GVF features that discretize the state variables into meaningful regions corresponding to an intuitive notion of safety. This includes two indicators for each variable corresponding to cart position, cart velocity, pole angle, and angle velocity. A perfectly balanced pole will always remain in the defined safe regions. These discretized features also double as the terminal features, since they capture the relevant aspects of the state at termination (being in a safe or unsafe region). For ESP-continuous, we have 12 features, the first 8 pre-terminal GVF features corresponding to the delta features of the cart position, cart velocity, pole angle, and angle velocity for left and right sides. The 4 terminal GVF features are indicators of whether the episode ended by moving out-of-bounds to the left or right or the pole fell below the termination agle to the left or right.

**Tug of War.** Tug of War (ToW) is an adversarial two-player strategy game we designed using PySC2 for Starcraft 2. ToW is interesting for humans and presents many challenges to RL including an enormous state space, thousands of actions, long horizons, and sparse reward (win/loss). A detailed description is in Appendix C. ToW is played on a rectangular map divided into top and bottom horizontal lanes. Each lane has two bases structures at opposite ends, one for each player. The first player to destroy one of the opponent's bases in either lane wins. The game proceeds in 30 second waves. By the beginning of each wave, players must decide on either the top or bottom lane, and how many of each type of military production building to purchase for that lane. Purchases are constrained by the player's available currency, which is given at a fixed amount each wave. Each purchased building produces one unit of the specified type at the beginning of each wave. The units move across the lanes toward the opponent, engage enemy units, and attack the enemy base if close enough. The three types of units are Marines, Immortals, and Banelings, which have a rock-paper-scissors relationship and have different costs. If no base is destroyed after 40 waves, the player with the lowest base health loses. In this work, we trained a single agent against a reasonably strong agent produced via pool-based self-play learning (similar to AlphaStar training(Vinyals et al., 2019)).

We present two ToW ESP agents that use 17 and 131 structured GVF features (noting that the 131 features are very sparse). These feature sets are detailed in Appendix E. For the 17 feature agent, the pre-terminal features correpsond to the delta damage to each of the four bases by each of the three types of units; allowing GVFs to predict the amount of base damage done by each type of unit, giving insight into the strategy. Note that there is no natural discretization of the numeric damage variables and hence we only consider the delta encoding. The terminal GVF features are indicators of which base has the lowest health at the end of the game and whether the game reached 40 waves. The terminal GVF features encode the possible ways that the game can end. The 131 feature agent extends these features to to keep track of damage done in each lane to and from each combination of unit types along with additional information about the economy.

## 6.2 LEARNING PERFORMANCE

To evaluate whether using ESP models hurts performance relative to "standard" models we compare against two DQN instances: *DQN-full* uses the same overall network architecture as ESP-DQN, i.e. the GVF network structure feeding into the combining network. However, unlike ESP-DQN, the DQN-full agent does not have access to GVF features and does not attempt to train the GVF network explicitly. It is possible DQN-full will suffer due to the bottleneck introduced at the interface between the GVF and combiner networks. Thus, we also evaluate *Vanilla DQN*, which only uses the combining network of ESP-DQN, but directly connects that network to the raw agent input. Details of network architectures, optimizers, and hyperparameters are in the Appendix D.

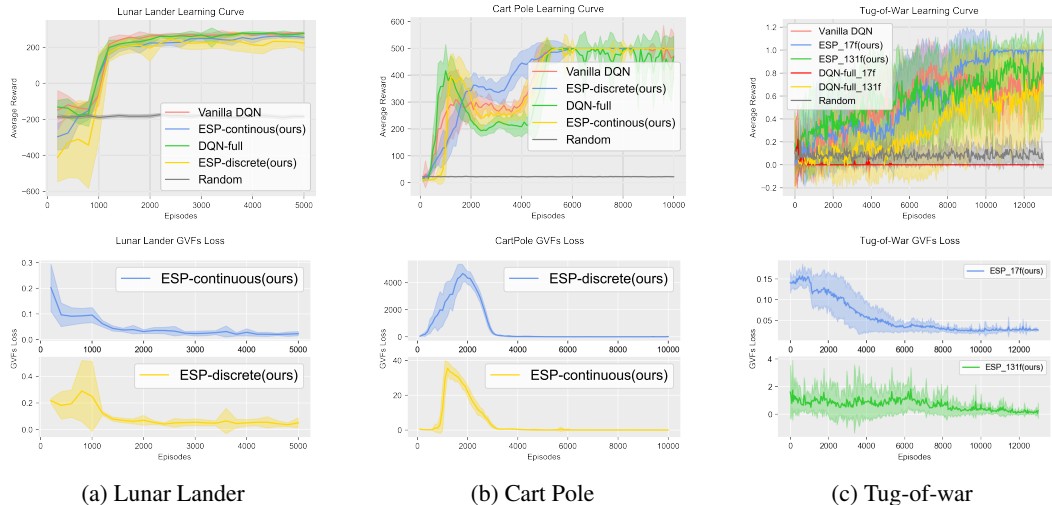

Figure 2: Reward learning curves (top row) and GVF Loss learning curves (bottom row) for the different agents in three environments. We show the mean +/- std over 10 independent runs.

Figure 2 (top row) shows the learning curves for different agents and for the random policy. All curves are averages of 10 full training runs from scratch using 10 random seeds. For the control problems, CartPole (with discrete and continous GVFs features) and LunarLander, we see that all agents are statistically indistinguishable near the end of learning and reach peak performance after about the same amount of experience. This indicates that the potential complications of training the ESP model did not significantly impact performance in these domains. We see that the discrete feature version of CartPole converged slightly faster than the continous version, but the difference is relatively small. For ToW, the ESP-DQN agents perform as well or better than the DQN variants, with all agents showing more variance. ESP-DQN with 17 features consistently converges to a win rate of nearly 100% and is more stable than the 131-feature version and other DQN variants. Interestingly, DQN-full with 17 features consistently fails to learn, which we hypothesize is due to the extreme 17 feature bottleneck inserted into the architecture. This is supported by seeing that with 131 features DQN-full does learn, though more slowly than ESP-DQN.

To evaluate the GVF accuracy of ESP-DQN we produce ground truth GVF data along the learning curves. Specifically, given the ESP policy $\hat{\pi}$ at any point, we can use Monte-Carlo simulation to estimate $Q_F^{\hat{\pi}}(s, a)$ for all actions at a test set of states generated by running $\hat{\pi}$. Figure 2 (bottom row) shows the mean squared GVF prediction error on the test sets as learning progresses. First, for each domain the GVF error is small at the end of learning and tends to rapidly decrease when the policy approaches its peak reward performance. LunarLander and ToW show a continual decrease of GVF error as learning progresses. CartPole, rather shows a sharp initial increase then sharp decrease. This is due to the initially bad policy always failing quickly, which trivializes GVF prediction. As the policy improves the GVFs become more challenging to predict leading to the initial error increase.

## 6.3 EXAMPLE EXPLANATIONS

Appendix F includes a larger set of examples with detailed analysis in each domain.

**Lunar Lander.** In Figure 3a, the game state (top) shows a state in Lunar Lander entered by a near-optimal learned ESP policy. The state is dangerous due to the fast downward and clockwise rotational velocity depicted by arrows. The GVFs (bottom-left) shows the Q-values for the actions and the predicted GVF bars. We see that the "main engine" and "right engine" actions have nearly the same Q-values with "main engine" slightly preferred, while "left engine" and "noop" are considered significantly worse. We want to understand the rationale for the strong and weak preferences.

While a user can observe differences among GVFs across actions, it is not clear how they relate to the reference. The IG and MSX (bottom-right) shows the IGXs corresponding to the preference of "main engine" over the other three actions. In addition, the MSX is depicted via dashed lines over IGX components in the MSX. Focusing first on the larger preferences, "main engine" is preferred to

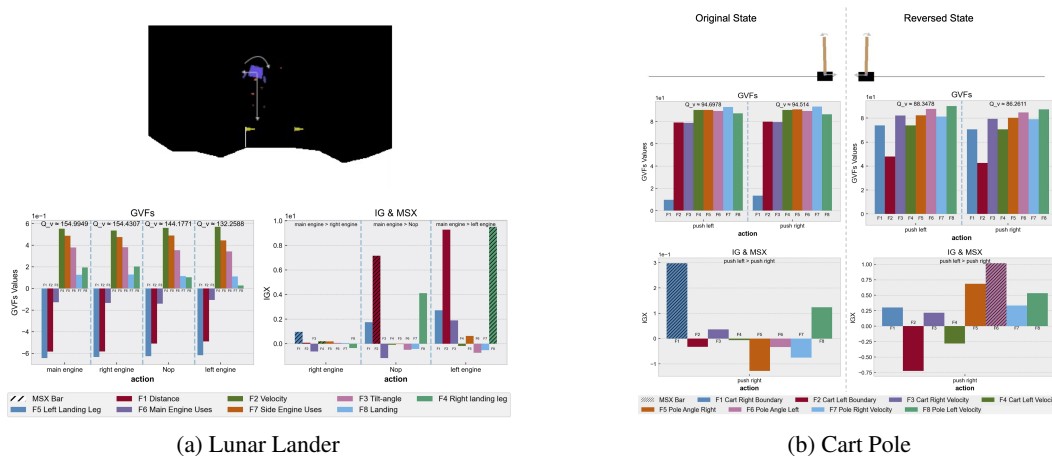

(a) Lunar Lander  (b) Cart Pole

Figure 3: Explanation examples for Lunar Lander (left) and CartPole (right). Each example shows the game state, the Q-values and GVF predictions for actions, and the IGX and MSX.

"left engine" primarily due to GVF differences in the velocity and landing features, with the MSX showing that landing alone is sufficient for the preference. This rationale agrees with common sense, since the left engine will accelerate the already dangerous clockwise rotation requiring more extreme actions that put the future reward related to landing at risk.

For the preference over "noop" the velocity feature dominates the IGX and is the only MSX feature. This agrees with intuition since by doing nothing the dangerous downward velocity will not be addressed, which means the landing velocity will have a more negative impact on reward. Comparing "main engine" to the nearly equally valued "right engine" shows that the slight preference is based on the distance and right leg landing feature. This is more arbitrary, but agrees with intuition since the right engine will both reduce the downward velocity and straighten the ship, but will increase the leftward velocity compared to the main engine. This puts it at greater risk of reducing reward for missing the right leg landing goal and distance reward. Overall the explanations agreed well with intuition, which together with similar confirmation can increase our confidence in the general reasoning of the policy. We also see the MSXs were uniformly very small.

**Cart Pole.** We compare a Cart Pole state-action explanation to an explanation produced by its reversed state as shown in Figure 3b. This comparison illustrates how in one case, the explanation agrees with intuition and builds confidence; while the other exposes an underlying inaccuracy or flaw.

Our original game state (left) positions the cart in a dangerous position moving right, close to the end of the track. The pole is almost vertical and has a small angle velocity towards the left. The action "push left" (move cart left) agrees with intuition as the cart is at the right edge of the screen and cannot move right without failing the scenario. The IG and MSX (left) concurs, showing the cart's current position close to the right edge as the main reason why it prefers the "push left" action over the "push right"; moving left will put the cart back within a safe boundary.

Reversing the game state (left) by multiplying -1 to each value in the input state vector produces a flipped game state (right). The cart is now positioned in a dangerous position moving left, close to the end of the track. Once again the pole is almost vertical and now has a small angle velocity towards the right. One would expect the agent to perform the action "push right" (the opposite action to game state (left)) as moving left will cause the agent to move off the screen and fail the scenario. However, as depicted in IG and MSX (right) we see the agent prefers "push left" over "push right". The agent justifies this action via an MSX that focuses on maintaining pole vertically to the left. This justification indicates that the agent is putting too much weight on the pole angle versus the boundary condition in this dangerous situation. The agent has not learned the critical importance of the left boundary. This indicates further training on the left side of the game map is needed. Presumably, during training the agent did not experience similar situations very often.

**Tug of War.** In Figure 4, we give 2 examples from a high-performing 17 feature ESP agent, one that agrees with common sense and one that reveals a flaw. Game state (top) shows the ESP agent (blue

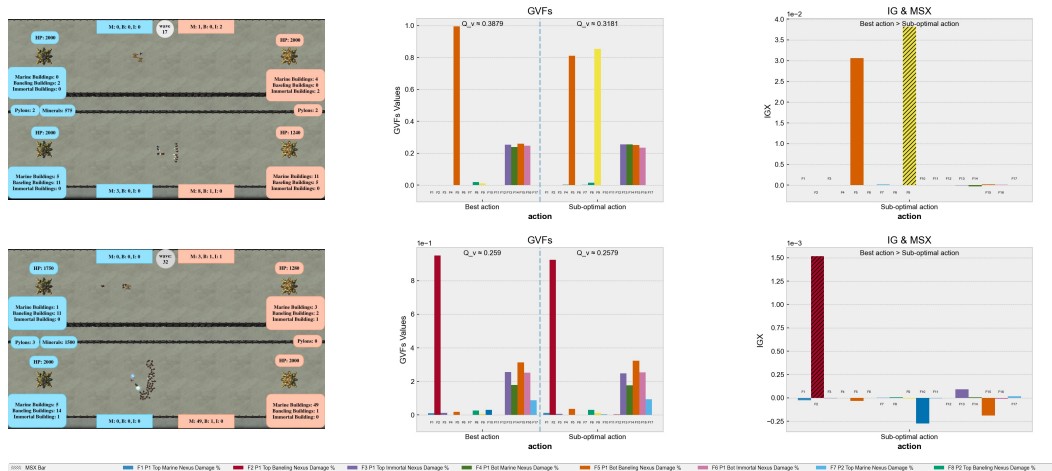

Figure 4: Example Explanations for Tug-of-War 17 feature ESP-DQN agent. Each row is a decision point showing: (left) game state; (middle) Q-values and GVFs for preferred action and a non-preferred action; (right) IGX for action pair and corresponding MSX (indicated by highlighted bars). For Game 1 (top) the agent's preferred action is +4 Marine, +1 Baneling in Top Lane and the non-preferred action is +10 Marine, +1 Baneling on Bottom. For Game 2 (bottom) the highest ranked action is +1 Baneling in Bottom Lane and sub-optimal action is +2 Marine, +4 Baneling in Bottom Lane.

player) with too few marine buildings to defend against the Immortals. We show information for the best ranked action and a sub-optimal action (details in caption). The best action creates top-lane units, while the sub-optimal action creates the maximum bottom-lane units. The IGX and MSX show (top) that the most responsible GVF feature for the preference is "damage to the top base from immortals", which agrees with intuition since the best action attempts to defend the top base, while the sub-optimal action does not. Indeed, the GVFs for the sub-optimal action show the top base is predicted to take 80% damage from the enemy's Immortals compared to nearly 0 for the best action.

In the second game state (bottom), the ESP agent plays against an opponent that it was not trained against and loses by having the bottom base destroyed. The state shows a large enemy attack in the bottom with the ESP agent having enough resources (1500 minerals) to defend if it takes the right action. However, the most preferred action is to add just one Baneling building to the bottom lane, which results in losing. Why was this mistake made?

We compare the preferred action to an action that adds more buildings to the bottom lane, which should be preferred. The IGX and MSX show that the action preference is dominated by the GVF feature related to inflicting damage in the top lane with Banelings. Thus, the agent is "planning" to save minerals to purchase more top Baneling buildings. The IGX does indicate that the agent understands the sub-optimal action will be able to defend the bottom lane, however, this advantage for the sub-optimal action is overtaken by the optimism about the top lane. This misjudgement of relative values causes the agent to lose the game. On further analysis, we found that this misjudgement is likely due to the ESP agent never experiencing a loss due to such a bottom lane attack during training.

## 7 SUMMARY

We introduced the ESP model for producing meaningful and sound contrastive explanations for RL agents. The key idea is to structure the agent's action-value function in terms of meaningful future predictions of its behavior. This allows for action-value differences to be compared in terms of deltas in the future behaviors they entail. To achieve meaningfulness, we required the agent designer to provide semantic features of the environment, upon which GVFs were learned. To achieve soundness, we ensured that our explanations were formally related to the agent's preferences in a well-defined way. Our case studies provide evidence that ESP models can be learned in non-trivial environments and that the explanations give insights into the agent's preferences. An interesting direction for future work is to continue to enhance the internal structure of the GVFs to allow for explanations at different levels of granularity, which may draw on ideas from GVF networks (Schlegel et al., 2018).

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

## A  ESP-DQN PSEUDO-CODE

The Pseudo-code for ESP-DQN is given in Algorithm 1.

---

**Algorithm 1 ESP-DQN:** Pseudo-code for ESP-DQN agent Learning.

---

**Require:** $\text{Act}(s, a)$ ;; returns tuple $(s', r, F, done)$ of next state $s'$, reward $r$, GVF features $F \in R^n$, and terminal state indicator $done$

**Require:** $K$ - target update interval, $\beta$ - reward discount factor, $\gamma$ - GVF discount factor

  Init $\hat{Q}_F, \hat{Q}'_F$ ;; The non-target and target GVF networks with parameters $\theta_F$ and $\theta'_F$ respectively.

  Init $\hat{C}, \hat{C}'$ ;; The non-target and target combining networks with $\theta_C$ and $\theta'_C$ rspectively.

  Init $M \leftarrow \emptyset$ ;; initialize replay buffer

  ;; Q-function is defined by $\hat{Q}(s, a) = \hat{C}(\hat{Q}_F(s, a))$

  ;; Target Q-function is defined by $\hat{Q}'(s, a) = \hat{C}'(\hat{Q}'_F(s, a))$

  **repeat**

    Environment Reset $s_0 \leftarrow$ Initial State totalUpdates $\leftarrow 0$

    **for** $t \leftarrow 0$ to $T$ **do**

      $a_t \leftarrow \epsilon(\hat{Q}, s_t)$ // $\epsilon$-greedy

      $(s_{t+1}, r_t, F_t, done_t) \leftarrow \text{Act}(s_t, a_t)$

      Add $(s_t, a_t, r_t, F_t, s_{t+1}, done_t)$ to $M$

      ;; update networks

      Randomly sample a mini-batch $\{(s_i, a_i, r_i, F_i, s'_i, done_i)\}$ from $M$

      $\hat{a}_i \leftarrow \arg\max_{a \in A} \hat{Q}'(s'_i, a)$

      $f'_i \leftarrow \begin{cases} F_i & \text{If } done_i \text{ is } \textbf{true} \\ F_i + \gamma \hat{Q}'_F(s'_i, \hat{a}_i) & \text{Otherwise} \end{cases}$

      $q'_i \leftarrow \begin{cases} r_i & \text{If } done_i \text{ is } \textbf{true} \\ r_i + \beta \hat{Q}'(s'_i, \hat{a}_i) & \text{Otherwise} \end{cases}$

      Update $\theta_F$ via gradient descent on average mini-batch loss $(f'_i - \hat{Q}_F(s_i, a_i))^2$

      Update $\theta_C$ via gradient descent on average mini-batch loss $(q'_i - \hat{Q}(s_i, a_i))^2$

      **if** totalUpdates mod $K == 0$ **then**

        $\theta'_F \leftarrow \theta_F$

        $\theta'_C \leftarrow \theta_C$

      **end if**

      totalUpdates $\leftarrow$ totalUpdates $+ 1$

      **if** $done_t$ is **true then**

        break

      **end if**

    **end for**

  **until** convergence

---

# B  CONVERGENCE PROOF FOR ESP-TABLE

Algorithm 2 gives the pseudo-code for ESP-Table based on $\epsilon$-greedy exploration. Note that, as for Q-learning, the convergence proof applies to any exploration strategy that guarantees all state-action pairs are visited infinitely often in the limit.

---

**Algorithm 2 ESP-Table:** Pseudo-code for a table-based variant of ESP-DQN. The notation $Q \xleftarrow{\alpha} x$ is shorthand for $Q \leftarrow (1 - \alpha)Q + \alpha x$.

---

**Require:** $\text{Act}(s, a)$ ;; returns tuple $(s', r, F)$ of next state $s'$, reward $r$, and GVF features $F \in R^n$
**Require:** $h(q)$ - hash function from $R^n$ to a finite set of indices $I$
**Require:** $K$ - target update interval
**Require:** $\gamma, \beta$ - discount factors for GVF and reward respectively
  Init $\alpha_{F,0}, \alpha_{F,0}$ ;; learning rates for GVF and combining function
  Init $\hat{Q}_F[s, a]$ ;; GVF table indexed by state-action pairs
  Init $\hat{C}[i]$ ;; Combining function table indexed by indices in $I$
  Init $\hat{Q}'_F[s, a]$ ;; Target GVF table indexed by state-action pairs
  Init $\hat{C}'[i]$ ;; Target Combining function table indexed by indices in $I$
  ;; Q-function is defined by $\hat{Q}(s, a) = \hat{C}[h(\hat{Q}_F[s, a])]$
  ;; Target Q-function is defined by $\hat{Q}'(s, a) = \hat{C}'[h(\hat{Q}'_F[s, a])]$

  $s_0 \leftarrow$ Initial State
  $t = 0$
  **repeat**
    **if** $t \bmod K == 0$ **then**
      $\hat{Q}'_F \leftarrow \hat{Q}_F$
      $\hat{C}' \leftarrow \hat{C}$
    **end if**
    $a_t \leftarrow \epsilon(\hat{Q}, s_t)$ ;; $\epsilon$-greedy exploration
    $(s_{t+1}, r_t, F_t) \leftarrow \text{Act}(s_t, a_t)$
    $a' \leftarrow \arg\max_a \hat{Q}'(s_{t+1}, a)$
    $\hat{Q}_F[s_t, a_t] \xleftarrow{\alpha_{F,t}} F_t + \gamma \hat{Q}'_F(s_{t+1}, a')$
    $\hat{C}[h(\hat{Q}_F[s_t, a_t])] \xleftarrow{\alpha_{C,t}} r_t + \beta \hat{Q}'(s_{t+1}, a')$
    $t \leftarrow t + 1$
  **until** convergence

---

For the proof we will let $t$ index the number of learning updates and $i = \lfloor t/K \rfloor$ be the number of updates to the target tables. The formal statements refer to the *"conditions for the almost surely convergence of standard Q-learing"*. These condition are: 1) There must be an unbounded number of updates for each state-action pair, and 2) The learning rate schedule $\alpha_t$ must satisfy $\sum_t \alpha_t = \infty$ and $\sum_t \alpha_t^2 < \infty$. ESP-Table uses two learning rates, one for the GVF and one for the combining function.

We will view the algorithm as proceeding through a sequence of *target intervals*, indexed by $i$, with each interval having $K$ updates. We will let $\hat{C}'_i$ and $\hat{Q}'_{F,i}$ denote the target GVF and combining functions, respectively, for target interval $i$ with corresponding target Q-function $\hat{Q}'_i(s, a) = \hat{C}'_i[h(\hat{Q}'_{F,i}[s, a])]$ and greedy policy $\hat{\pi}'_i(s) = \arg\max_a \hat{Q}'(s, a)$. The following lemma relates the targets via the Bellman backup operators. Below for a GVF $Q_F$ we define the max-norm as $|Q_F|_\infty = \max_s \max_a \max_k |Q_{f_k}(s, a)|$.

**Lemma 1.** *If ESP-Table is run under the standard conditions for the almost surely (a.s.) convergence of Q-learning and uses a Bellman-sufficient pair $(F, h)$ with locally consistent $h$, then for any $\epsilon > 0$ there exists a finite target update interval $K$, such that, with probability 1, for all $i$,*
$$\left| \hat{Q}'_{i+1} - B[\hat{Q}'_i] \right|_\infty \leq \epsilon \text{ and } \left| \hat{Q}'_{F,i+1} - B_F^{\hat{\pi}'_i}[\hat{Q}'_{F,i}] \right|_\infty \leq \epsilon.$$

That is, after a finite number of learning steps during an interval, the updated target Q-function and GVF are guaranteed to be close to the Bellman backups of the previous target Q-function and GVF.

Note that since the targets are arbitrary on the first iteration, these conditions hold for any table-based ESP Q-function.

*Proof.* Consider an arbitrary iteration $i$ with target functions $\hat{Q}'_i$, $\hat{C}'_i$, $\hat{Q}'_{F,i}$, and let $\hat{Q}^t_i$, $\hat{C}^t_i$, and $\hat{Q}^t_{F,i}$ be the corresponding non-target functions after $t$ updates during the interval. Note that for $t = 0$ the non-targets equal to the targets. The primary technical issue is that $\hat{C}^t_i$ is based on a table that can change whenever $\hat{Q}^t_{F,i}$ changes. Thus, the proof strategy is to first show a convergence condition for $\hat{Q}^t_{F,i}$ that implies the table for $\hat{C}^t_i$ will no longer change, which will then lead to the convergence of $\hat{C}^t_i$.

Each update of $\hat{Q}^t_{F,i}$ is based on a fixed target policy $\hat{\pi}'_i$ and a fixed target GVF $\hat{Q}'_{F,i}$ so that the series of updates can be viewed as a stochastic approximation algorithm for estimating the result of a single Bellman GVF backup given by

$$B_F^{\hat{\pi}'_i}[\hat{Q}'_{F,i}](s,a) = F(s,a) + \gamma \sum_{s'} T(s,a,s') \cdot \hat{Q}'_{F,i}[s', \hat{\pi}'_i(s')], \tag{2}$$

which is just the expectation of $F(s,a) + \gamma \hat{Q}'_{F,i}[S', \hat{\pi}'_i(S')]$ with $S' \sim T(s,a,\cdot)$. Given the conditions on the learning rate $\alpha_t$ it is well known that $\hat{Q}^t_{F,i}$ will thus converge almost surely (a.s.) to this expectation, i.e. to $B_F^{\hat{\pi}'_i}[\hat{Q}'_{F,i}]$.[1] The a.s. convergence of $\hat{Q}^t_{F,i}$ implies that for any $\epsilon'$ there is a finite $t_1$ such that for all $t > t_1$, $|\hat{Q}^t_{F,i} - B_F^{\hat{\pi}'_i}[\hat{Q}'_{F,i}]| \leq \epsilon'$. This satisfies the second consequence of the lemma if $\epsilon' \leq \epsilon$ and $K > t_1$.

Let $\epsilon' < \epsilon$ be such that it satisfies the local consistency condition of $h$, which implies that for all $t > t_1$ and all $(s,a)$, $h(\hat{Q}^t_{F,i}[s,a]) = h(B_F^{\hat{\pi}'_i}[\hat{Q}'_{F,i}[s,a]])$. That is, after $t_1$ updates, $h$ will map the non-target GVF to the same table entry as the Bellman GVF Backup of the target GVF and policy. Combining this with the Bellman sufficiency of $(F, h)$ implies that for any state action pairs $(s,a)$ and $(x,y)$, if $h(\hat{Q}^t_{F,i}[s,a]) = h(\hat{Q}^t_{F,i}[x,y])$ then $B[\hat{Q}'_i](s,a) = B[\hat{Q}'_i](x,y)$. This means that after $t_1$ updates all of the updates to a table entry $h(\hat{Q}^t_{F,i}[s,a])$ have the same expected value $B[\hat{Q}'_i](s,a)$. Using a similar argument as above this implies that $\hat{Q}^t_i(s,a) = h(\hat{Q}^t_{F,i}[s,a])$ converges a.s. to $B[\hat{Q}'_i](s,a)$ for all $(s,a)$ pairs. Let $t_2$ be the implied finite number of updates after $t_1$ where the error is within $\epsilon$. The target update interval $K = t_1 + t_2$ satisfies both conditions of the lemma, which completes the proof. □

Using Lemma 1 we can prove the main convergence result.

**Theorem 2.** *If ESP-Table is run under the standard conditions for the almost surely (a.s.) convergence of Q-learning and uses a Bellman-sufficient pair $(F, h)$ with locally consistent $h$, then for any $\epsilon > 0$ there exists a finite target update interval $K$, such that for all $s$ and $a$, $\hat{\pi}^t(s)$ converges a.s. to $\pi^*(s)$ and $\lim_{t \to \infty} |\hat{Q}^t_F(s,a) - Q_F^{\pi^*}(s,a)| \leq \epsilon$ with probability 1.*

*Proof.* From Lemma 1 we can view ESP-Table as performing approximate Q-value iteration, with respect to the sequence of target functions $\hat{Q}'_i$. That is, the total updates done during a target interval define an approximate Bellman backup operator $\hat{B}$, such that $\hat{Q}'_{i+1} = \hat{B}[\hat{Q}'_i]$. Specifically, there exists a $K$, such that the approximate operator is $\epsilon$-accurate, in the sense that for any Q-function $Q$, $\left| \hat{B}[Q] - B[Q] \right|_\infty \leq \epsilon$.

Let $\hat{B}^i[Q]$ denote $i$ applications of the operator starting at $Q$ so that $\hat{\pi}'_i$ is the greedy policy with respect to $\hat{B}^i[\hat{Q}'_0]$. Prior work (Bertsekas & Tsitsiklis, 1996) implies that for any starting $Q$, the sub-optimality of this greedy policy is bounded in the limit.

$$\limsup_{i \to \infty} \left| V^* - V^{\hat{\pi}'_i} \right|_\infty \leq \frac{2\beta}{(1-\beta)^2} \epsilon \tag{3}$$

---

[1] An "MDP-centric" way of seeing this is to view the update as doing policy evaluation in an MDP with discount factor 0 and stochastic reward function $R(s,a,s') = F(s,a) + \gamma \hat{Q}'_{F,i}(s', \hat{\pi}'_i(s'))$. The convergence of policy evaluation updates then implies our result.

where $V^*$ is the optimal value function and $V^\pi$ is the value function of a policy $\pi$.

Now let

$$\delta = \min_\pi \min_{s:V^*(s) \neq V^\pi(s)} |V^*(s) - V^\pi(s)|$$

be smallest non-zero difference between an optimal value at a state and sub-optimal value of a state across all non-optimal policies. From this definition it follows that, if $\left| V^* - V^{\hat{\pi}^i} \right|_\infty \leq \delta$, then $\hat{\pi}^i = \pi^*$. From Equation 3 this condition is achieved in the limit as $i \to \infty$ if we select $\epsilon < \frac{(1-\beta)^2}{2\beta} \delta$. Let $K_1$ be the finite target interval implied by Lemma 1 to achieve this constraint on $\epsilon$. Since Lemma 1 holds with probability 1, we have proven that $\hat{\pi}'_i$ converges almost surely to $\pi^*$ for a finite $K_1$. This implies the first part of the theorem.

For the second part of the theorem, similar to the above reasoning, Lemma 1 says that we can view the target GVF $\hat{Q}'_{F,i}$ as being updated by an approximate Bellman GVF operator $\hat{B}_F^{\hat{\pi}'_i}$. That is, for any GVF $Q_F$ and policy $\pi$, $\left| B_F^\pi[Q_F] - \hat{B}_F^\pi[Q_F] \right|_\infty \leq \epsilon$. Further, it is straightforward to show that our approximate Bellman GVF operator satisfies an analogous condition to Equation 3, but for GVF evaluation accuracy in the limit. In particular, for any $\pi$ and initial $Q_F$, if we define $\hat{Q}_{F,i}^\pi$ to be the GVF that results after $i$ approximate backups the following holds.[2]

$$\limsup_{i\to\infty} \left| Q_F^\pi - \hat{Q}_{F,i}^\pi \right|_\infty \leq \frac{\epsilon}{(1-\gamma)}. \tag{4}$$

Thus, for a fixed policy the approximate backup can be made arbitrarily accurate for small enough $\epsilon$.

From the almost sure convergence of $\hat{\pi}'_i$, we can infer that there exists a finite $i^*$ such that for all $i > i^*$, $\hat{\pi}'_i = \pi^*$. Thus, if $K > K_1$, then after the $i^*$ target update the target policy will be optimal thereafter. At this point the algorithm enters a pure policy evaluation mode for fixed policy $\pi^*$, which means that the approximate GVF operator is continually being applied to $\pi^*$ across target intervals. From Equation 4 this means that in the limit as $i \to \infty$ we have that

$$\limsup_{i\to\infty} \left| Q_F^{\pi^*} - \hat{Q}'_{F,i} \right|_\infty \leq \frac{\epsilon}{(1-\gamma)}.$$

Thus, we can achieve any desired accuracy tolerance in the limit by selecting a small enough $\epsilon$. Let $K_2$ be the target interval size implied by Lemma 1 for that epsilon and let the target interval be $K = \max\{K_1, K_2\}$. This implies that using a target interval $K$, there is a finite number of target updates $i'$ after the first $i^*$ updates such that for all $i > i^* + i'$, $\hat{Q}'_{F,i}$ will achieve the error tolerance. This completes the second part of the proof. □

## C    TUG OF WAR DOMAIN

In this section, we overview the real-time strategy (RTS) game, 'Tug of War' (ToW), used for this study. Tug of War (ToW) is an adversarial two-player zero-sum strategy game we designed using Blizzard's PySC2 interface to Starcraft 2. Tug of War is played on a rectangular map divided horizontally into top and bottom lanes as shown in Figure 5. The game is viewed from an omnipotent camera position looking down at the map. Each lane has two base structures; Player 1 owns the two bases on the left of the map, and Player 2 owns the two bases on the right. The game proceeds in 30 second waves. Before the next wave begins, players may select either the top or bottom lane for which to purchase some number of military-unit production buildings with their available currency.

We have designed Tug of War allowing AI vs AI, Human vs Human, and AI vs Human gameplay. Watch a Human vs Human ToW game from Player 1's perspective here: `https://www.youtube.com/watch?v=krfDz0xjfKg`

Each purchased building produces one unit of the specified type at the beginning of each wave. Buildings have different costs and will require players to budget their capital. These three unit types,

---

[2]This can be proved via induction on the number of exact and approximate Bellman GVF backups, showing that after $i$ backups the difference is at most $\epsilon \sum_{j=0}^{i-1} \gamma^j$ and then taking the limit as $i \to \infty$.

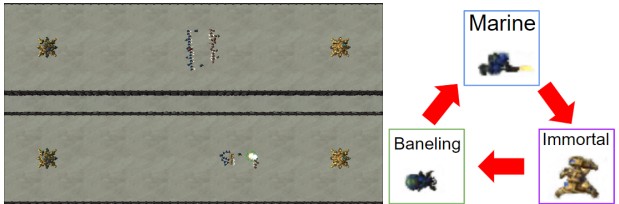

Figure 5: (left) Tug of War game map - Top lane and bottom lane, Player 1 owns the two bases on the left (gold star-shaped buildings), Player 2 owns the two bases on the right. Troops from opposing players automatically march towards their opponent's side of the map and attack the closest enemy in their lane. (right) Unit Rock Paper Scissors - Marines beats Immortals, Immortals beats Banelings, and Banelings beats Marines. We have adjusted unit stats in our custom Starcraft 2 map to befit ToW's balance.

Marines, Immortals, and Banelings, have strengths and weaknesses that form a rock-paper-scissors relationship as shown in Figure 5. Units automatically move across the lanes toward the opponent's side, engage enemy units, and attack the enemy base if close enough. Units will only attack enemy troops and bases in their lane. If no base is destroyed after 40 waves, the player who owns the base with the lowest health loses.

Both Players receive a small amount of currency at the beginning of each wave. A player can linearly increase this stipend by saving to purchase up to three expensive economic buildings, referred to as a Pylon.

ToW is a near full-information game; players can see the all units and buildings up to the current wave. Both player's last purchased buildings are revealed the moment after a wave spawns. The only hidden information is the unspent currency the opponent has saved; one could deduce this value as the wave number, cost of each building, currency earned per wave, and the quantities of buildings up to the current snapshot are known. It would be difficult for a human to perform this calculation quickly.

Tug of War is a stochastic domain where there is slight randomness in how opposing units fight and significant uncertainty to how the opponent will play. Winning requires players assessing the current state of the game and balancing their economic investment between producing units immediately or saving for the future. Players must always be mindful of what their opponent may do so as to not fall behind economically or in unit production. Purchasing a Pylon will increase one's currency income and gradually allow the player to purchase more buildings, but players must be wary as Pylons are expensive, saving currency means not purchasing unit-production buildings which may lead to a vulnerable position.Conversely, if the opponent seems to be saving their currency, the player can only guess as to what their opponent is saving for; the opponent may be saving to purchase a Pylon or they may be planning to purchase a lot of units in a single lane.

Tug of War presents a challenging domain to solve with Reinforcement Learning (RL). These challenges include a large state space, large action space, and sparse reward. States in ToW can have conceivably infinite combinations of units on the field, different quantities of buildings in lanes, or different base health. The number of possible actions in a state corresponds to the number of ways to allocate the current budget, which can range from 10s to 1000s. Finally, the reward is sparse giving +1 (winning) or 0 (losing) at the end of the game, where games can last up to 40 waves/decisions.

## C.1    TUG OF WAR FEATURE DESIGN

While humans need continuous visual feedback to interact with video games, computer systems can use simple numeric values received in disjointed intervals to interpret game state changes. We have designed an abstract "snapshot" of the ToW game state at a single point in time represented as a 68 dimensional feature vector. Note that for this study, we have increased added additional features to capture granular details, thus bringing the total to 131 features. At the last moment before a wave spawns, the AI agent receives this feature snapshot and uses it to select an action for the next wave. We call this moment a decision point. The decision point is the only time when the agent receives information about the game and executes an action; the agent does not continuously sample

observations from the game. The agent's performance indicates this abstraction is sufficient for it to learn and play the game competently.

The state feature vector includes information such as the current wave number, health of all 4 bases, the agent's current unspent currency, the agent's current building counts in both top and bottom lanes, the enemy's last observed building counts in the top and bottom lanes, pylon quantities, and the number of troops in each grid of the 4 grid sections of the map as depicted in Figure 6. opponent's current unspent mineral count is not sent to the agent as this hidden information is part of the game's design.

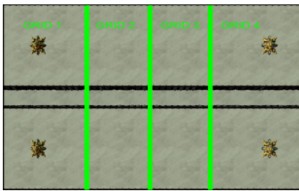

Figure 6: ToW 2 Lane 4 Grid - Unit quantities and positions on the map is descretized into four sections per lane.

## D   AGENT DETAILS: HYPERPARAMETERS AND ARCHITECTURES

The ESP agent code is provided in Supplementary Material, including pre-trained models for all domains we present.

Table 1 gives the hyperparameters used in our implementation. Note that our implementation of ESP-DQN supports both hard target updates as shown in the pseudo-code and "soft target updates" (Lillicrap et al., 2015), where at each step the target network parameters are gradually moved toward the currently learned parameters via a mixing proportion $\tau$. We found that this can sometimes lead to more stable learning and use it in two of our domains as indicated in the table.

Table 2 presents our GVF network structures used to train the agents in each domain. The choice of activation function for each GVF output component is based on the output type of the component. For GVF outputs in $[0, 1]$ we use a sigmoid activation, for sets of mutually exclusive indicator GVFs (e.g. the win condition) we use a softmax activation over the set, and for GVF outputs with arbitrary numeric ranges we use a linear activation. Specifcially, we use Sigmoid functions on F1 through F12 and F17 features for our Tug of War ESP-DQN 17-feature agent and on F131 for our Tug of War ESP-DQN 131-feature agent because the data ranges $(0, 1)$. We apply a SoftMax function to features F13 to F16 and F1 to F8 for our our Tug of War ESP-DQN 17-feature and 131-feature agents because said features correspond to probabilities that sum to 1.

## E   TUG OF WAR 131 FEATURES

We introduce a detailed description of the 131 features used to train our Tug of War ESP-DQN agent. These features capture events in ToW, namely:

- Game ending win-condition probabilities; The likely-hood for each base to be destroyed or have the lowest HP at wave 40.
- P1 and P2 currency; These features allow GVFs to predict the amount of money players will receive in the future.
- Quantity of units spawned.
- The number of each type of units will be survive at different ranges [4] we defined on the map for both players; allowing the GVFs to predict the advantage of each lane of each type of unit in the future.

---

[4]In addition to the 4 grid map regions as explained in Figure 6, we add a 5th map region (Grid 5) to detect units attacking bases. Grid 5 for P1 is indicates the quantity of P1 units attacking P2's bases. This is reversed for P2, where now Grid 1 for P2 indicates P2 units attacking P1's bases.

| Hyper-Parameters | Lunar Lander | Cart Pole | Tug-of-War(both) |
|---|---|---|---|
| Discount factors($\gamma$ and $\beta$) | 0.99 | 0.99 | 0.9999 |
| Learning Rate($\alpha$) | $10^{-4}$ | $10^{-5}$ | $10^{-4}$ |
| Start Exploration($\epsilon_s$) | 1.0 | 1.0 | 1.0 |
| Final Exploration($\epsilon_f$) | 0.01 | 0.05 | 0.1 |
| Exploration Decrease(linearly) Steps | $2*10^5$ | $2*10^5$ | $4*10^4$ |
| Batch Size | 128 | 128 | 32 |
| Soft/Hard Replace | Soft | Soft | Hard |
| Soft Replace($\tau$) | $5*10^{-4}$ | $5*10^{-4}$ | N/A |
| Hard Replace Steps | N/A | N/A | $6*10^3$ |
| GVF Net Optimizer | Adam | Adam | Adam |
| Combiner Net Optimizer | SGD | SGD | SGD |
| Training Episodes | $5*10^3$ | $10^4$ | $1.3*10^4$ |
| Evaluation Intervals | 200 | 100 | 40 |
| Evaluation Episodes | 100 | 100 | 10 |
| Riemann approximation steps of IGX[3] | 30 | 30 | 30 |

Table 1: Hyper-parameters and optimizers used to train our ESP-DQN and DQN agents on Lunar Lander, Cart Pole and Tug of War.

| | Lunar Lander | Cart Pole | Tug-of-War(17f) | Tug-of-War(131f) |
|---|---|---|---|---|
| GVF Net | 3 layers MLP | 3 layers MLP | 4 layers MLP | 4 layers MLP |
| GVF Output Activation Function | Linear | Linear | Sigmoid(F1-F12, F17) SoftMax(F13-F16) | SoftMax(F1-F8) Sigmoid(F131) |
| Combiner Net | 3 layers MLP | 3 layers MLP | 4 layers MLP | 4 layers MLP |

Table 2: Network structures we used to train our ESP-DQN and DQN agents on Lunar Lander, Cart Pole and Tug of War.

- Delta damage to each of the four bases by each of the three unit types. These features allow GVFs to predict the amount of damage each unit type will inflict on the opponent's base in the unit's respective lane.

- The amount of damage inflicted by which type of units on another type of units for both players, like the damage the friendly Marine inflicted on enemy immortal; Allows the GVFs to predict the amount damage for each type of units inflicting on each type of units.

- An indicator of whether the game reaches waves of tie-breaker.

## F   EXAMPLE EXPLANATIONS

**Cart Pole.** Figure 7a shows a Cart Pole state encountered by a learned near-optimal ESP policy, where the cart and the pole are moving in the left direction with the pole angle being in a dangerous range already. The action "push left" is preferred over "push right", which agrees with intuition. We still wish to verify that the reasons for the preference agree with our common sense. From the IGX and MSX in Figure 7c the primary reason for the preference is the "pole angle left" GVF, which indicates that pushing to the right will lead to a future where the pole angle spends more time in the dangerous left region. Interestingly we see that "push right" is considered advantageous compared to "push left" with respect to the left boundary and left velocity features, which indicates some risk for push left with respect to these components. All of these preference reasons agree with intuition and along with similar examples can build our confidence in the agent.

**Lunar Lander.** Figure 8a illustrates a Lunar Lander state achieved by a near-optimal ESP policy. The lander is moving down to the left and is close to landing within the goal. Additionally, the left leg has touched the ground as marked by the green dot. Figure 8b shows the GVF values of all actions and expects the lander to land successfully with the right leg touching down. The GVFs values of the distance, velocity, and angle are small because the lander is close to the goal.

Although this state allows the lander to successfully land after taking any action, the IGX shown in Figure 8c illustrates the agent prefers "use left engine". This is because using the right engine will

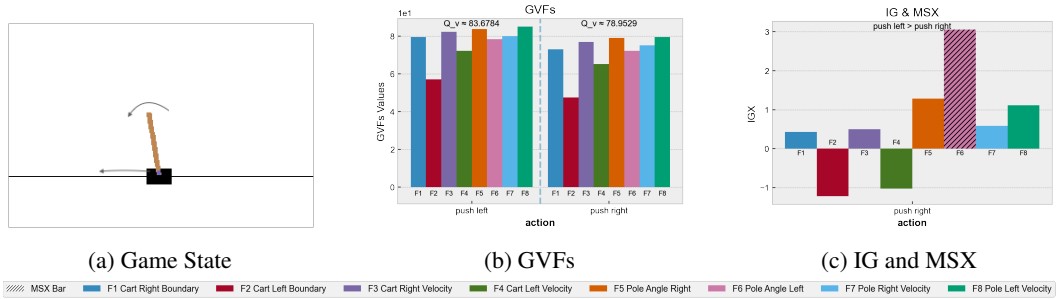

Figure 7: Explanation example for Cart Pole. Three Figures show the game state, the Q-valuesand GVF predictions for actions, and the IGX and MSX respectively.

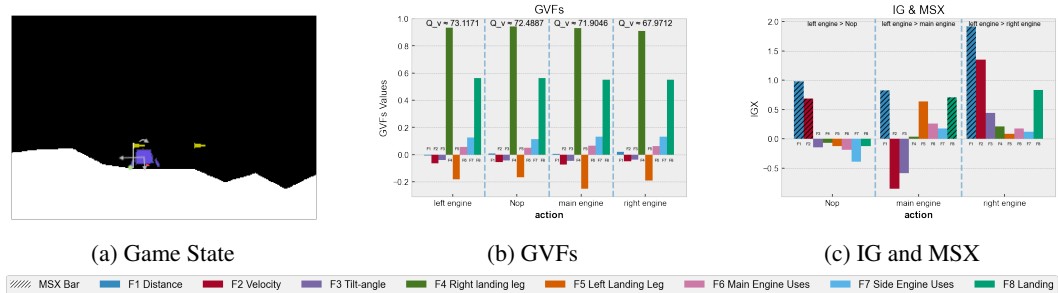

Figure 8: Explanation example for Lunar Lander. Three Figures show the game state, the Q-values and GVF predictions for actions, and the IGX and MSX respectively.

increase the velocity of the lander, pushing it towards the left and increasing "F1 Distance" from the goal. This action and justification makes intuitive sense as the lander is unlikely to fail in this state and has chosen an action that reduces its velocity and decreases its landing delay.

The "use main engine" action also delays the landing increases distance to the goal, as indicated by the MSX bars in Figure 8c. The IGX also shows the "use main engine" engine risks the left leg leaving the ground which agrees with intuition as moving up pushes the lander back into space. However, "use main engine" gives the lander another opportunity to adjust its velocity and angle. That may be why the IGX of velocity and tile-angle are negative. The "no-op" action has a lower preference than the best action because the lander is slightly drifting and may move out of the goal. Two largest IGXs of "noop" action agrees with this rationale. However, the IGX of landing is negative that may be arbitrary or indicates doing the "noop" action will lead the lander to land faster since the lander is moving down already, but sometimes landing faster gets less reward because moving to the center of the goal can gain more reward by reducing the distance between the center of goal and lander.

**Tug of War: 17 Features.** Figure 9a depicts a screenshot of a Tug of War game where our ESP agent (P1, blue) is playing against a new AI opponent (P2, orange) it has never encountered. The ESP agent's top base is destroyed after two waves thus losing the game. The annotated game state shows the ESP agent doesn't have enough units to defend its top base as its opponent's banelings can kill almost all its units, and the agent's Top lane base has approximately 35% hit points (HP) remaining. We can regard this state as a critical moment in the game because the agent spends all its money to defend the top lane and still looses the base in two waves after taking the its highest ranked action. Given our deep ToW game knowledge, we want to understand why the ESP agent chose to purchase Banelings in the Bottom Lane (arguably sub-optimal) rather than purchase Immortals in the Top Lane (intuitively a better action).

To understand why the agent prefers the action that is worse than an action we intuitively recognize to be better, we analyze both action's GVFs (Figure 9b), and IG & MSX (Figure 9c). The sub-optimal action's GVFs shows the sub-optimal action is expected to reduce damage from the enemy's top Banelings. This indicates the agent understands taking the sub-optimal action can pose a better

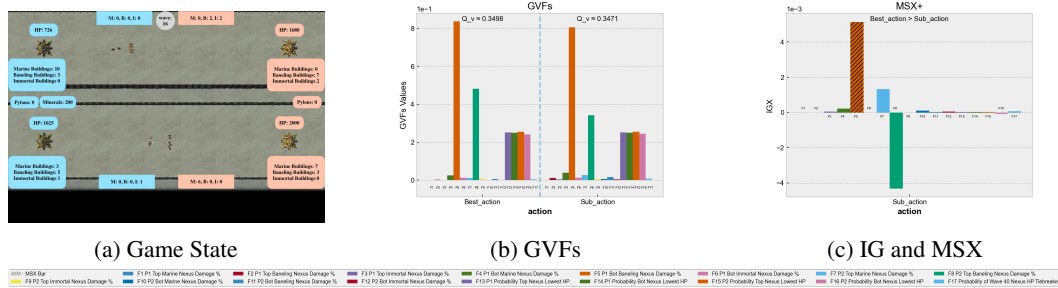

| (a) Game State | (b) GVFs | (c) IG and MSX |

Figure 9: Explanation example for Tug-of-War 17 feature ESP-DQN agent. Three Figures show the game state, the Q-values and GVF predictions for actions, and the IGX and MSX respectively. The top ranked action +2 Baneling in Bottom Lane and sub-optimal is +1 Immortals in Top Lane

defense. However, the MSX bar shows positive IGX of the self bottom Baneling damage still can cover the negative IGX of enemy top Baneling damage; indicating the agent is focusing on destroying the enemy's bottom base while ignoring the damage its top base will take. This misjudgement can be attributed to the agent over-fitting to its fixed-agent opponent during training.

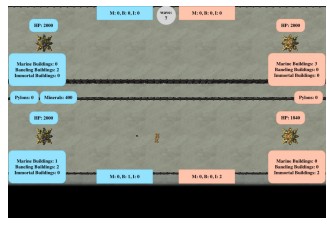

(a) Game State

**Tug of War: 131 Features.** Figure 10a depicts a screenshot of a Tug of War game where our (P1, blue) ESP agent is playing against the same fixed-policy AI opponent (P2, orange) it was trained against. The ESP agent wins by destroying the opponent's bottom base. The state in Figure 10a indicates both players have a balanced quantity of units in the top lane. We also observe P2 has an advantage in the bottom lane as the ESP agent doesn't have enough units to defend. The ESP agent has determined its best action is to spend all its money on producing +8 Marine buildings in the Bottom Lane to defend, which agrees with intuition as Marines counter Immortals. To justify why one can regard this choice as optimal, we compare the agent's best-determined action, +8 Marine buildings in Bottom Lane, to a sub-optimally ranked action, +5 Baneling buildings in Bottom Lane, due to Immortals counter Banelings.

Figure 10b shows the GVF value of both action. Given the dense nature of the 131 Features, we summarize the following:

- The values concerning accumulated quantity features such as future currency to be earned are higher in the sub-optimal action than the best action because the game is expected to be prolonged if a sub-optimal action is taken. The probability to end the game by tie-breaker(F131) as shown in Figure 10b, graph "Probability to End by Tie-breaker" agrees taking the best action leads to a faster win.

- The sub-optimal action raises the probability of our ESP agent's bottom base getting destroyed(F2) and lowers the probability of the opponent's bottom base getting destroyed(F4). This assessment agrees with the game rules as Banelings do little to counter Immortals.

- Agent's Expected Bottom Marine to Spawn(F14) is higher of if it takes the best action, and Expected Bottom Baneling to Spawn(F15) is higher if takes sub-optimal action.

- By taken the best action, the agent expects its future surviving bottom marines to be closer to P2's bottom base(F44, F47 and F50); indicating the agent's units are able to push the enemy back. Contrasted to the sub-optimal action, where the opponent's surviving bottom Immortal

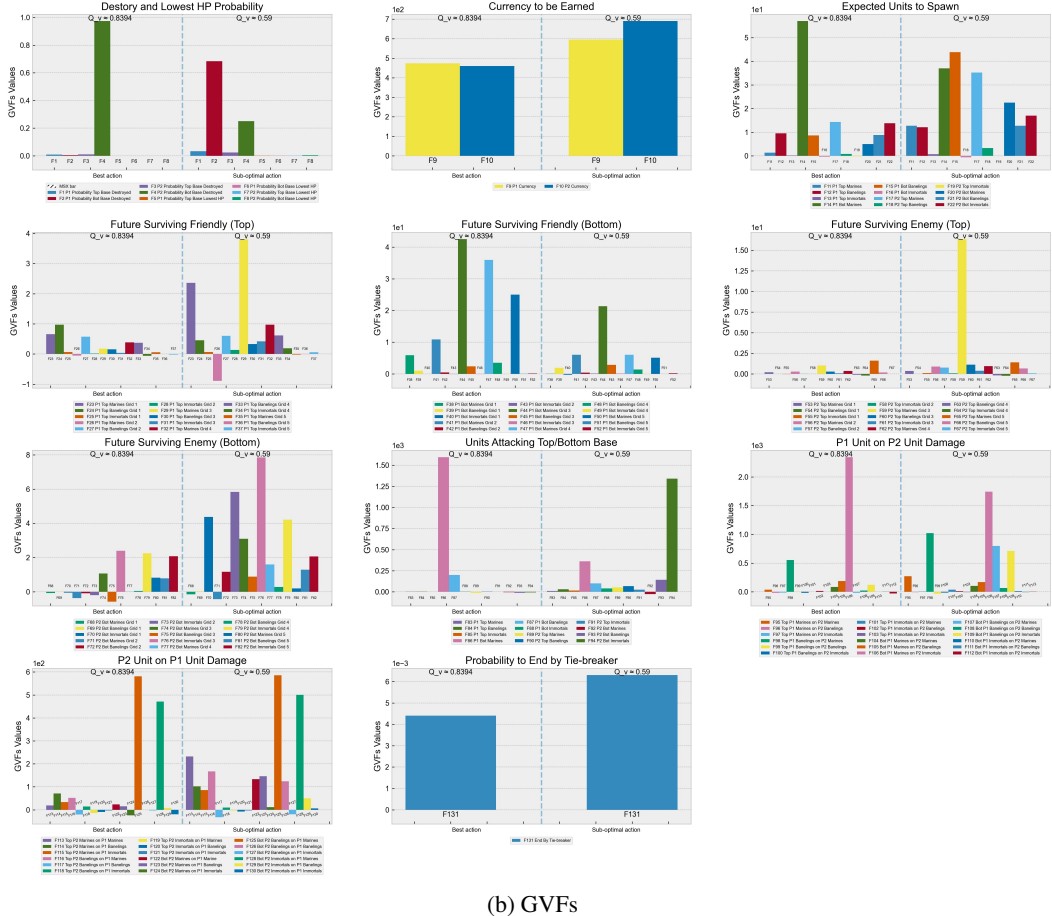

(b) GVFs

is expected to be closer to the ESP agent's bottom base(F70, F73 and F76), indicating the opponent pushed the agent back.

- If the ESP agent purchases +8 marines in the bottom lane (best ranked action), the agent expects to take no damage from the enemy(F89 to F94). This can be contrasted to the expected damage if the agent were to purchase +5 baneling buildings in the bottom lane (sub-optimal action) where the agent expects to take base damage from P2's immortals(F94) as shown in Figure 10b, graph "Units Attacking Top/Bottom Base".

- We can validate the agent understands the rock-paper-scissor interaction between marines, banelings, and immortals from the GVF graphs as shown in Figure 10b, graph "P1 Unit on P2 Damage" and "P2 Unit on P1 Damage". If the agent produces marines, the ESP agent correctly expects to inflict a large amount of damage on P2's immortals. If the agent produces banelings, the ESP agent correctly expects to inflict a large amount of damage on P2's marines.

- There exist some flaws in the agent's GVF predictions. Some values such as Future Surviving Units in Figure 10b should not be negative, indicating some flaw in the agent's training. This suggests an engineer can add a ReLU function on the output to prevent negative values.

Explanations produced by our ESP model are sound because said explanations do not depend on GVF comparisons alone. Figure 10c, graph "Units Attacking Top/Bottom Base" illustrates P2 Immortal Damage on bottom base (F94); the primary MSX contribution for why the agent ranked +8 marine buildings as its best action. Given the notion that P2's Immortals in the bottom lane presents a significant threat, producing marines to defend the immortals makes good intuitive sense. Banelings are a sub-optimal choice in this scenario, and would do little to defend against Immortals. We summarize the IG and MSX graph in Figure 10c as follows,

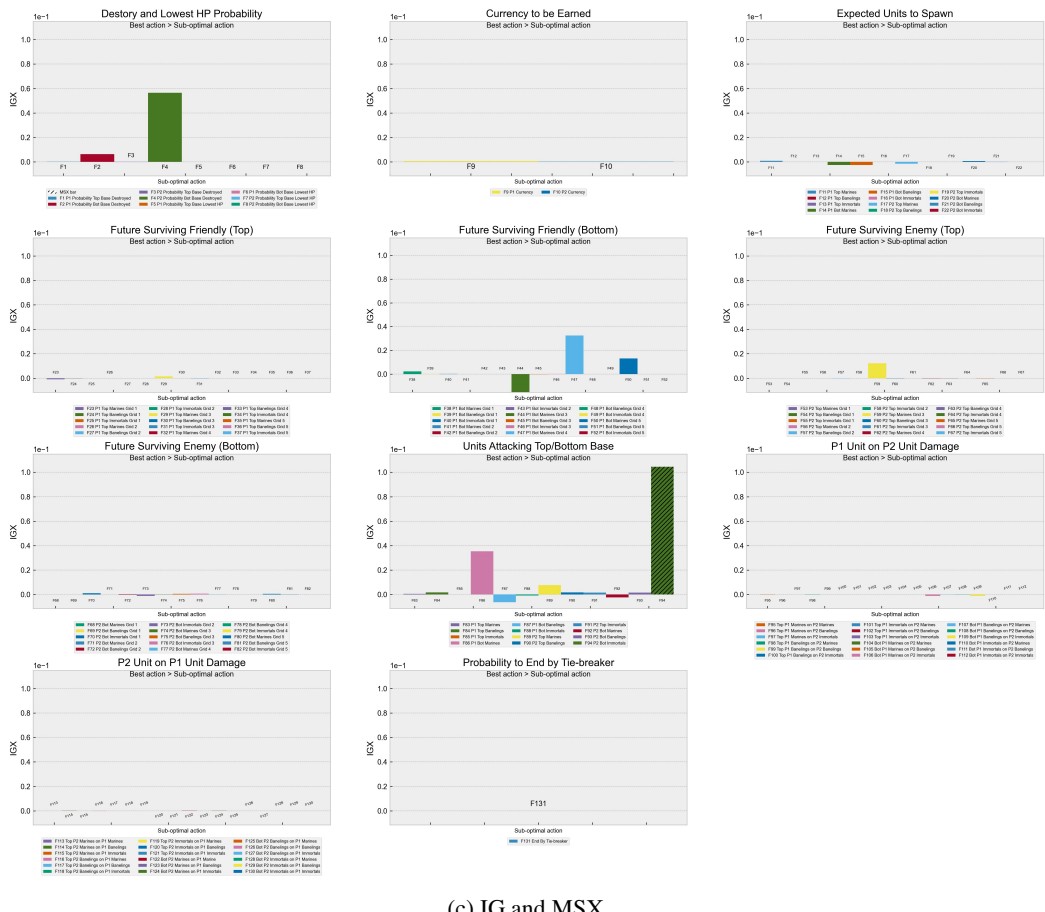

(c) IG and MSX

Figure 10: Explanation example for Tug-of-War 131 feature ESP-DQN agent. Since there are too much features to show as one figure, we separate them into 11 clusters. Three Figures show the game state, the Q-values and GVF predictions for actions, and the IGX and MSX respectively. The top ranked action +8 Marines in Bottom Lane and sub-optimal is +5 Banelings in Bottom Lane.

- The best action adds more Marine buildings; thus increasing the quantity of marines spawned per wave, but the agent doesn't care about the quantity of marines(F14) as the IGX is close to 0. However, the agent cares about the damage the Marine inflict (F86), although this is not as important as defending against opponent's Immortals.

- Graph "Destroy and Lowest HP Probability" illustrates the two mutually exclusive win types in ToW; winning by destroying one of P2's bases, or winning by making sure one of P2's base has the lowest HP at wave 40. The probability Base Destroyed IGX indicates the agent expects to destroy the opponent's bottom base(F4) and defend its own bottom base(F2).

- Graph "Future Surviving Friendly (Bottom)" illustrates the contribution of P1's surviving troops in the bottom lane. The positive IGX contribution of feature "P1 Bottom Marine Grid 4(F47)" and "Grid 5(F50)" indicates the agent cares about its marines moving closer to the enemy's bottom base. The IGX of the "P1 Bot Marine Grid 3(F44)" is negative, possibly because Grid 3 is too far from the opponent's base to be considered a disadvantage.

Given the large number of features, the MSX is critical to get a quick understanding of the agent's preference. In general, user interface design will be an important consideration when the number of features is large. Such interfaces should allow users to incrementally explore the IGX and GVFs of different actions flexibly and on demand.

