# OpenReview forum: "Contrastive Explanations for Reinforcement Learning via Embedded Self Predictions"
_ICLR.cc/2021/Conference — ICLR 2021 Oral_

### Official Review · AnonReviewer3 · 2020-10-28
**ESP appears to rely on careful GVF feature engineering; further information/evaluation needed**

**Rating:** 7
**Confidence:** 5

**Review:**

Edit:

I have read the authors' response as well as the other reviews. Based on the additional results and added feature selection details, I now agree that ESP is generally applicable.

Summary:

The authors present ESP, an RL system that can then explain action choices in terms of future feature values. Generalized value functions (GVFs) are used to learn an estimate of total future discounted feature values. These estimated total feature values are then used to estimate each actions Q-value. Since Q-values are based on GVF outputs, these intermediate values can be used as an explanation. The authors select a subset of these values to form a Minimal Sufficient Explanation (MSX). The proposed system is evaluated using three domains. The authors show that performance is comparable to non-GVF agents.

Reasons for score:

Though the authors present a novel explanation format, the applicability of the method is uncertain. The results appear to rely on specific GVF feature choices. Non-general methods are still of interest, but the limited information on feature construction prevents a fair comparison to other approaches. Additionally, the explanations are not evaluated quantitatively.

Pros:

-The use of GVFs for explanations in terms of future feature values is a novel line of work. MSXs are a natural way to then produce more concise explanations, and the authors extend MSX to their non-linear use case in a well-justified way.

-The analysis of the Tug of War explanations was thorough. It clearly showed how ESP explanations would be used to investigate agent behavior.

Cons:

-ESP is built upon the GVF features, but the choice of GVF features is suspect. Each environment uses a different style for its features. Lunar Lander and CartPole both have continuous features, yet the authors use "deltas" for Lunar Lander and region discretization for Cart Pole. Tug of War uses a bunch of features, including information about feature values when a game ends, and non-linearities are applied to the outputs of the GVF features. Note that these non-linearities are used *only* for Tug of War, and different non-linearities are used for different features (Table 2 in Appendix D). Effectively, each environment appears to use carefully engineered features. Given that DQN-Full performs very poorly for specific settings (i.e., in some cases, the agent cannot learn without the GVF features), the choice of features seems to be important. The authors should indicate the process used for selecting them and how these features should be chosen for other environments. ESP may not be robust to GVF feature choice, but this is insufficiently addressed in the paper.

-In Section 6.3, the authors present potential conclusions that can be drawn from an ESP agent. These conclusions can be evaluated to determine whether valid conclusions can be drawn from the explanations. Such an evaluation would allow the hypotheses of the authors to be tested.

-A substantial reorganization of the paper would improve clarity. Various definitions and descriptions are provided a few sections after the terms/methods are first used. Terms are unnecessarily over-loaded (such as "sound").

Questions During Rebuttal Period:

-Please address and clarify the "Cons" above.

-In particular:

a) How were the GVF features chosen? Why does each environment use different features?

b) What happens when Lunar Lander is given "CartPole-style" (region discretization) features? What happens when CartPole is given "Lunar-Lander-style" (change in features) features?

Minor comments:

-This work would benefit from another editing pass for tense/plurality matching between subject and verb.

-The proofs in Appendix B would benefit from a re-write; currently, they are hard to parse.

---

> ### Author Response · Authors · 2020-11-25
> **Response to Reviewer 3**
>
> Thanks for the suggestions. We have incorporated them into the revised paper. In the general comment to all reviewers we have included a change log for the updated paper.
>
> “non-linearities are applied to the outputs of the GVF features. Note that these non-linearities are used only for Tug of War, and different non-linearities are used for different features (Table 2 in Appendix D).”
>
> Re: Here we assume you are referring to the activation functions used to make the final prediction of the GVF outputs. Our choices of activation functions here are standard and based on only the data type the GVF output---no clever engineering is being used. Specifically, we use logistic outputs for predicting GVFs that will lie between [0,1], we use softmax over sets of GVF features that are mutually exclusive indicators  (e.g. the type of game outcome), and a pure linear/identity activation function when the output has a larger or unspecified real-valued range. This was partly discussed in the paragraph before Table 1 in Appendix D in the original submission. The revised paper describes the above in the last paragraph of Appendix D.
>
> “Effectively, each environment appears to use carefully engineered features.”
>
> Re: We hope that the discussion below in response to your next concern will change your view here. There was very little cleverness involved in selecting the features for these domains. However, we don’t want to argue that in general everyone should avoid human ingenuity when designing features for important applications. That is one of the places where humans+ML can be quite synergistic.
>
> “The authors should indicate the process used for selecting them and how these features should be chosen for other environments.”
>
> Re: This was a great suggestion. We included at the start of Section 6.1 a full paragraph on the general schema we followed for selecting the features in our environments, which could be followed in other environments. To summarize that discussion, there are two types of features: 1) terminal features that describe how an episode ends, which are often very useful for explanations, 2) non-terminal features that describe how the state evolves. In our case, these features are completely based on environment variables that describe the state and reward components. For (2) we suggest two general approaches based on either discretization and deltas. Which of those choices is best can depend on the domain and intuition about the interpretability value.
>
> We have extended the experiments (Figure 2) to show that the CartPole ESP model can be learned with both types of features (deltas and discretization). We have also done this for LunarLander to illustrate learnability. However, for LundarLander there is not an obvious semantic type of discretization so we used a generic one using 8 uniform bins. In practice, we would not use this because it is not useful for interpretability, but here we at least illustrate that learning can be successful.
>
> “ These conclusions can be evaluated to determine whether valid conclusions can be drawn from the explanations.”
>
> Re: This is an interesting suggestion that requires some thought about how to carry out properly when it is possible. We just haven’t had time to do this for the revised version of the paper, for which, we focused mostly on addressing your feature engineering concerns by running additional experiments. Note that the loss of the GVFs is very low as shown in the graphs based on empirical evaluations. So that aspect of the conclusion is already tested in a strong sense. Identifying the rare corner cases that correspond to potential weaknesses suggested by the conclusions is the part that requires more effort and we will look into this for the final version.
>
> “a) How were the GVF features chosen? Why does each environment use different features?”
> b) What happens when Lunar Lander is given "CartPole-style" (region discretization) features? What happens when CartPole is given "Lunar-Lander-style" (change in features) features?”
>
> Re: We hope this was addressed sufficiently by adding the features design paragraph and running LunarLander and CartPole on both types of features to show that the learning approach can work in both cases (details above and in Figure 2). It just turned out that in the original submission we chose the feature type in each domain that was intuitively judged to be better for interpretability.
>
> “-The proofs in Appendix B would benefit from a re-write; currently, they are hard to parse.”
> Re: We haven’t had time to revisit that section, but will do so based on feedback from colleagues for the final version. If there are particular aspects that are unclear, we are happy to incorporate that feedback.

---

### Official Review · AnonReviewer1 · 2020-10-28
**Good submission**

**Rating:** 8
**Confidence:** 4

**Review:**

### Summary

The paper attempts to improve the interpretability of RL agents' action selection process by (a) proposing embedded self-prediction (ESP), a model that embeds generalised value functions (GVFs) in the action-value function of the agent with a "combining" function to and (b) ESP-DQN, an extension of DQN that augments experience replay tuples with a GVF feature vector and that decomposes the model into separate combining and GVF parameters. These enable to define action-values with respect to predefined feature maps, thus providing more "resolution" into the behaviour of the policy.

### Good stuff

1. The idea of decomposing the policy into GVFs as a way to force explanations wrt. some features is *brilliant*, and it is well executed when combined with the contrastive explanation system.
2. Sections 2-4 provide both a clear introduction to GVFs as well as a detailed and sound description of the overall framework.
3. The related work section is fairly tight, but actually covers a good amount of necessary and relevant related work, which makes it easy to scan through the literature.
4. The experimental section does employ mostly fairly similar environments, but it is clear in the hypotheses being tested, and it is fairly satisfactory considering what it is attempting to evaluate.

### Uncategorised notes

- This is more of a meta-comment, but I enjoyed reading the paragraph about manually-designed features in Section 1. I understand why the authors felt the need to write it, and it is a sad state of affairs that it is now often a requirement to argue what to many people is just reality.
- I wonder if it'd be worth it to test the method on Atari, through possibly the use of MinAtar: https://github.com/kenjyoung/MinAtar -- It feels like there's a gap in difficulty (of learning and analysis) between the ToW setting and the rest of the environments, and something aking to a middle ground would probably be a good setting to add. Complex gridworlds such as BabyAI and the NetHack Learning Environment are probably also good options.

### Final comments

Overall, I'm extremely happy to strongly recommend this paper towards acceptance. It is well written, it introduces a method that attempts to move forward towards solving an important problem in Reinforcement Learning, and there's a significant amount of details in the paper that would make it fairly straightforward to reproduce.

### Typos
#### Sec 1

- RL agents explain its... -> their
- directly "embed"... -> embeds
- train those... -> trains

#### Sec 3
- Combination function update -> Combining? The manuscript is at times a little inconsistent.

---

> ### Author Response · Authors · 2020-11-25
> **Response to Reviewer 2**
>
> Thanks for positive feedback and suggestions. In the general comment to all reviewers we have included a change log for the updated paper.
>
> “I understand why the authors felt the need to write it, and it is a sad state of affairs that it is now often a requirement to argue what to many people is just reality.”
>
> Re: Hopefully the need for this type of paragraph is only short term. It just seems that the excitement about deep learning allowing for less feature engineering (replaced by architecture engineering) caused a strong blanket resistance to any feature engineering by some researchers. Hopefully this will balance out with time.
>
> “I wonder if it'd be worth it to test the method on Atari, through possibly the use of MinAtar:”
>
> Re: Thanks for the pointer. This looks like a very compelling evaluation option. We don’t have experience with it yet, but are definitely considering it for some followup work with this model and later extensions.

---

### Official Review · AnonReviewer2 · 2020-10-30
**Interesting paper at the intersection of RL and interpretability**

**Rating:** 7
**Confidence:** 2

**Review:**

This paper proposes a method that offers explanation of action preference in a deep RL agent based on given features by the human. In other words, the model explains why action A is preferred to action B based on some given features. This is done through the embedded self-prediction (ESP) model. The authors also proposed a method for evaluation of importance of features in the learned policy. While the paper benefits from extended experimental result and interesting theoretical analysis in the tabular RL, I think its readability could be improved. For example, I believe generalized value functions should be explained more extensively as (I think) it is less known to the community compared to concepts like MDP and DQN. Also, an analysis  (or at least some discussion) on the effect of the number of features on learning QFs would be helpful. Another question I have, is about the dependence of features. What would happen to the evaluation if some given features are dependent?
Some minor points:
- It would be helpful if the authors add a figure of their networks similar to what is common in the field, specifying input and output of the networks.
- The phrase of "greedy policy" was a little confusing for me, especially because "greedy algorithm" is usually one-step look ahead search.  Is it just argmax Q(s, a) (as suggested in page 3)?
- I think the defense on manually-designed features could be transferred to conclusion since the importance of interpretability has been already mentioned in the first paragraph.

---

> ### Author Response · Authors · 2020-11-25
> **Response to Reviewer 1**
>
> Thanks for positive feedback and suggestions. In the general comment to all reviewers we have included a change log for the updated paper.
>
> “Also, an analysis (or at least some discussion) on the effect of the number of features on learning QFs would be helpful.”
>
> Re: We have added a short discussion (last paragraph Section 2). It indicates that the framework can always work with just a single-feature, just the reward function, but that doesn’t provide any additional interpretability. In general, the set of features needs to have enough meaningful features so that the GVFs are sufficient for representing the Q-function, while supporting meaningful decomposition for contrastive explanation.
>
> “What would happen to the evaluation if some given features are dependent?”
>
> Re: As long as the features are sufficient for representing the Q-function, dependent features do not cause a problem in terms of learning the Q-function. For example, in our Tug-of-War agent with 131 features, there are a number of dependent ones. The more dependent the features are, however, the more possible ways there are to combine their GVFs for a Q-function approximation. The explanations, of course, will reflect the particular way those features are combined.
>
> “It would be helpful if the authors add a figure of their networks similar to what is common in the field, specifying input and output of the networks.”
>
> Re: We added Figure 1, which is referenced when introducing the ESP model in Section 2.
>
> “The phrase of ‘greedy policy’ was a little confusing for me, especially because "greedy algorithm" is usually one-step look ahead search. Is it just argmax Q(s, a) (as suggested in page 3)?”
>
> Re: Yes. We have updated the text in the introduction and second paragraph of Section 2 to clarify that the greedy policy is the ‘Q-function maximizing greedy policy’.
>
> “I think the defense on manually-designed features could be transferred to conclusion since the importance of interpretability has been already mentioned in the first paragraph.”
>
> Re: The intention is for it to be less missable by including it in the introduction. We’ve found that using such features to support interpretability is surprisingly controversial, which is why we wanted to make the argument more prominent. As reviewer 2 states, it is too bad that this type of argument is necessary.

---

### Author Response · Authors · 2020-11-25
**Change Log**

Thanks for all of the suggestions for improving the paper. We uploaded a revised paper that includes improvements based on those suggestions. Below is a change log and we have also referenced specific changes in the responses to each reviewer.

Change Log:
1. Updated the Cart Pole learning curves and GVF loss curves (Figure 2) by adding a result of the agent with continuous “deltas” features.

2. Added a new paragraph which is called "Schema for Selecting GVF Features" at the start of Section 6.1 to describe the schema we used and that could be used in other domains.

3. Rewrote the description of each environment so that description of features aligns with (2) above.

4. Added a diagram of the ESP model (Figure 1)

6. Updated the text in the introduction and second paragraph of Section 2 to clarify that the greedy policy is the ‘Q-function maximizing greedy policy’.

7. Added a short extended discussion of the influence of feature choice at the end of Section 2 (last paragraph).

8. Added an expanded discussion of the activation functions used in the GVF network architecture (last paragraph Appendix D).

---

### Decision · Program_Chairs · 2021-01-07
**Final Decision**

**Decision:**

Accept (Oral)

**Comment:**

This paper tackles the important problem of endowing deep RL agents with added interpretability. Action values are decomposed as the combination of GVFs learned on externally-specified features, offering action explanations in terms of discounted future returns in the space of interpretable quantities. Reviewers praised the approach, as well as the level of detail for reproducibility purposes. R3 had concerns about the generality of the method but follow-up experiments have allayed these concerns. Given the reviewer response and the central importance of the problem considered to the field, I can wholeheartedly recommend acceptance.